# Calibration by Air in Polarization Sensing

**Sergei N. Volkov [1],\*, Ignatii V. Samokhvalov [2] and Duk-Hyeon Kim [3]**

[1]  Center of Laser Atmosphere Sensing, V.E. Zuev Institute of Atmospheric Optics SB RAS, 1, Academician Zuev Square, 634055 Tomsk, Russia
[2]  Department of Optoelectronic Systems and Remote Sensing National Research, Tomsk State University, 36, Lenin Ave., 634050 Tomsk, Russia; lidar@mail.tsu.ru
[3]  School of Basic Science, Hanbat National University, Yuseong-gu, Daejeon 34158, Korea; dhkim7575@hanmail.net
\*  Correspondence: srgy_volkov@yahoo.com

**Abstract:** Scattered light polarization serves as an indicator and a characteristic of various processes in the atmosphere. The polarization measurements of all scattering matrix elements provide an adequate description of the optical and morphological parameters and orientation of particles in clouds. In this article, we consider the problem of the calibration of matrix polarization lidar (MPL) parameters. Calibration by air is an effective alternative to the technique for correcting optical element parameters and among the calibration parameters of the MPL optical path are the relative transmission coefficient of a two-channel receiver, the angular positions of the transmission axes of the optical elements of the transmitter and receiver units, including the polarizers and wave plates, and the retardance of wave plates. For the first time, the method of calibration by air was partially implemented in the MPL to study Asian dust in the atmosphere. We considered the calibration problem more generally and this was due to the need to calibrate different MPL modifications from stationary to mobile ones. The calibration equations have been derived in terms of instrumental vectors, and the method of their solution by the generalized least squares method has been proposed. The method has been verified on a numerical MPL model and validated using MPL measurements in Daejeon, Republic of Korea.

**Keywords:** polarization lidar; backscattering; Mueller matrix; wave plate



## 1. Introduction

Laser polarimetric remote sensing is successfully used to study the optical characteristics of the atmosphere [1–7]. The widespread polarization sensing method consists in measuring the cross-polarized components of irradiance scattered along the sensing path; however, within the limits of representation of the particle scattering using the Stokes parameters, this is only a part of the information on the scattering matrix elements. Complete information on the particle scattering matrix is obtained from measurements with a matrix polarization lidar (MPL) [8–11].

As it was shown at the presentation [12] at the 28th International Conference, 4–8 July 2022, Tomsk, the calibration of the matrix polarization lidar parameters is an urgent problem for remote sensing of the atmosphere. According to the existing practice, the lidar parameters are estimated using the technique of correcting the optical element parameters [8–10,13–16]; however, this correction method is pertinent only for laboratory studies. For mobile lidars operating in an autonomous mode in a wide range of external conditions, the calibration by measurements in the atmosphere becomes the main method.

The procedure of calibration by air has been partially implemented for the first time in the MPL to study Asian dust in the atmosphere [11]. For calibration, a segment of the sensing path with a prevalence of molecular scattering was chosen. In a measurement series, four wave plates were alternately replaced in the optical lidar channel. As a result of

calibration, the relative transmission coefficient of the two-channel receiver (below referred to as the relative transmission coefficient) and the angular positions of the transmission axes of the wave plates and polarization beam splitter (PBS) were estimated.

The calibration problem has been extensively reformulated and its solution has been significantly revised after MPL modernization. The lidar design was simplified using the procedure of an angular rotation of one wave plate of the laser radiation transmitter and another wave plate of the scattered radiation receiver. The calibration segment was chosen in air, and a series of measurements were carried out. The relative transmission coefficient in the optical channel of the MPL receiver was estimated by means of constructing and solving a linear system of equations. The angular positions of the transmission axes of the optical elements and retardance wave plates were estimated by solving the system of the nonlinear equations with the Gauss–Newton method.

The derivation of the calibration equation in terms of instrumental vectors is presented. A two-stage method for estimating the parameters from the obtained calibration equations with the generalized least squares method using available information is also described.

Using statistical methods, the stability of the method, rate of its convergence, deviations of estimates, and behavior for weak echo-signals were checked. For this purpose, a series of echo-signals for several signal levels were modeled. These data were then used to solve the inverse problem of a parameter reconstruction. The scalability of the method for a set of calibration parameters then allowed the method to be validated using the results of the MPL polarization sensing in Daejeon, Republic of Korea [11].

In Section 2 of the present work, the calibration problem is formulated and the principles of constructing the calibration equations for lidar echo signals are described. In Section 3, methods for solving linear and nonlinear calibration problems are presented. In Section 4, the procedure is described and the results of the verification and validation of the method are given. In the Appendix A, the order of representation and the calculated partial derivatives of the instrumental vectors are presented.

## 2. Calibration Equations

Here, we mainly use the notations of [11,17]. In particular, electromagnetic waves are described in the (x, y, z) coordinate system in which the z axis is parallel to the radiation propagation direction. In addition, we follow the traditional agreement for the light beam passing through an ideal optical element: the direction of the rotation of vibrational ellipse axes is positive for the clockwise rotation in the direction of the radiation source. If we designate the vector basis of the optical element by $(\mathbf{e}_{II}, \mathbf{e}_{\perp}, \mathbf{e}_z)$, we consider that the vector $\mathbf{e}_{II}$ defines the horizontal transmission axis relative to the y axis, and the direction of the vector $\mathbf{e}_z$ coincides with the z axis of the incident light beam.

Assume that the radiation source and the receiver of the scattered radiation of the MPL are located in one place. Then, for the scattering volume $\Delta V$ of the particles located at the distance $h$ from the receiver, the Stokes vectors of incident, $\mathbf{S}_0 = (I, Q, U, V)^T$, and scattered radiation, $\mathbf{S}$, are related by the formula [8]:

$$\mathbf{S} = \left(\Delta V/h^2\right)\mathbf{M}_\pi\mathbf{S}_0 \tag{1}$$

where the elements of the backscatter matrix $\mathbf{M}_\pi$, in $m^{-1}sr^{-1}$, are the volume backscattering coefficients and $T$ is the transposition index. This is the first order approximation of the scattering theory.

The matrix $\mathbf{M}_\pi$ can be represented as the sum $\mathbf{M}_\pi = \mathbf{A}_\pi + \mathbf{\Sigma}_\pi$ of the scattering matrices of particles, $\mathbf{A}_\pi$, and the molecular atmosphere, $\mathbf{\Sigma}_\pi$. With an allowance for depolarization,

the backscattering matrix in air, $\boldsymbol{\sigma}_\pi = \boldsymbol{\Sigma}_\pi / \boldsymbol{\Sigma}_{11}$, in which the element $\boldsymbol{\Sigma}_{11}$ is normalized by unity, has the form [18]:

$$\boldsymbol{\sigma}_\pi = \begin{pmatrix} 1 & 0 & 0 & 0 \\ 0 & 0.97 & 0 & 0 \\ 0 & 0 & -0.97 & 0 \\ 0 & 0 & 0 & -0.94 \end{pmatrix} \tag{2}$$

The form of the matrix $\boldsymbol{\sigma}_\pi$ is well known; therefore, it is natural to calibrate the MPL parameters by measuring the radiation scattering characteristics in air. The simplified optical scheme of the MPL is shown in Figure 1. Laser radiation is linearly polarized in the yoz scattering plane, and the irradiance $I$ is normalized by unity. Then, the normalized Stokes vector of the laser radiation is $\mathbf{s}_0 = (1, q, u, v)^T = (1, 1, 0, 0)^T$.

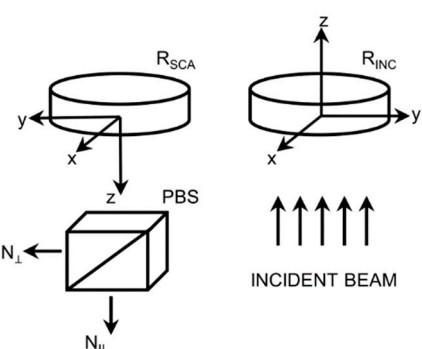

**Figure 1.** Optical scheme of the matrix polarization lidar. Here, 'incident beam' denotes the laser radiation, $R_{INC}$ is the transmitter wave plate, $R_{SCA}$ is the receiver wave plate, PBS is the polarization beam splitter, and $N_{II}$ and $N_\perp$ are the orthogonal recorded signals. The receiving collimator and the recording system are not shown here.

The receiver coordinate system is obtained via rotation through an angle of 180° in the scattering plane. At first, the beam is incident on the wave plate $R_{INC}$ with the transmission axis at an angle of $\phi_{INC}$ to the scattering plane, and is scattered in the atmosphere. In the receiver, the scattered radiation passes through the wave plate $R_{SCA}$ with the transmission axis at an angle of $\phi_{SCA}$ to the scattering plane. Then, the beam is split into two orthogonal components with the help of the polarization beam splitter (PBS) and is recorded as the signals $N_{II}$ and $N_\perp$. The receiving collimator and the recording system are not shown in the figure.

Consider the MPL calibration based on the measurements of scattered light irradiance as a series of echo signals $\{N_{II}, N_\perp\}$ also designated as signals. Each $i$th measurement in the series is made after rotations of the transmitter wave plate $R_{INC}$ and the receiver wave plate $R_{SCA}$. These are the rotations of the wave plates around the z axis in a series of angular states $\{\phi_{INC}, \phi_{SCA}\}$. The calibration is aimed at determining the systematic deviations of the MPL parameters: the relative transmission coefficient $\alpha$; angular positions of the transmission axes; wave plate retardances $\Delta\varphi_{INC}$, $\Delta\delta_{INC}$, $\Delta\varphi_{SCA}$, and $\Delta\delta_{SCA}$; the angle $\xi$ of the polarizer transmission axis.

Suppose that the signals $N_{II}$ and $N_\perp$ from the scattering layer $\Delta h$ at the distance $h$ are recorded in the photoelectron counting mode. Omitting for simplicity the serial number of the measurement, we consider the polarization sensing equations in the first approximation of multiple scattering theory as [8,11]:

$$N_{II} = NI^{II}_{SCA} \boldsymbol{\sigma}_\pi I_{INC} \tag{3}$$

$$N_\perp = \gamma N I_{SCA}^\perp \boldsymbol{\sigma}_\pi I_{INC} \tag{4}$$

Here, $\gamma$ is the coefficient due to the difference in the transmission of the channel optics and the photomultiplier sensitivity; $N = \eta N_0 \Delta h A h^{-2} \Sigma_{11} T^2(h)$; $\eta$ is the loss factor; $N_0$ is the number of photons per laser pulse; $A$ is the effective receiver aperture; $T^2(h)$ is the atmospheric transmission factor; $I_{SCA}^{II} = \mathbf{g}_0 \mathbf{M}_{II} \mathbf{M}_{SCA}$ and $I_{SCA}^\perp = \mathbf{g}_0 \mathbf{M}_\perp \mathbf{M}_{SCA}$ are the instrumental vectors of the receiver row; $I_{INC} = \mathbf{M}_{INC} \mathbf{s}_0$ is the instrumental vector of the transmitter column; $\mathbf{g}_0 = (1,\, 0,\, 0,\, 0)$ is the unit row vector; $\mathbf{M}_{II/\perp}$ are the PBS Mueller matrices [16], where $\mathbf{M}_{SCA/INC}$ are the Mueller matrices of the receiver and transmitter wave plates given by Equation (A5). We consider that hereinafter, the signal relationships in the form of the equations hold true with the accuracy determined by measurement statistics.

As shown in Ref. [11], if we define the polarization ratio as $c = (N_{II} - \alpha N_\perp)/(N_{II} + \alpha N_\perp)$, where $\alpha = 1/\gamma$, after a substitution of Equations (3) and (4), we obtain the MPL calibration equation:

$$I_{SCA} \boldsymbol{\sigma}_\pi I_{INC} = 0 \tag{5}$$

where the transmitter and receiver instrumental vectors $I_{INC}$ and $I_{SCA}$, using the terminology of Ref. [11], are given by Equations (A1) and (A2). With am=n allowance for the Equations (2), (A1) and (A2), Equation (5) is reduced to the form:

$$f_J = f_0 - c = 0 \tag{6}$$

where:

$$f_0 = 0.97 q_{INC} q_{SCA} - 0.97 u_{INC} u_{SCA} - 0.94 v_{INC} v_{SCA} \tag{7}$$

Here, $f_J \equiv f_J(\alpha, \varphi_{INC}, \delta_{INC}, \varphi_{SCA}, \delta_{SCA}, \xi)$ is the nonlinear functional of the calibration parameters except for $\alpha$.

Note that the matrix $\boldsymbol{\sigma}_\pi$ is diagonal; therefore, the sum of the signals $N_{II} + \alpha N_\perp$ is independent of the angular positions of the PBS and receiver and transmitter wave plates; therefore:

$$N_{II} + \alpha N_\perp = N \left( I_{SCA}^{II} + I_{SCA}^\perp \right) \boldsymbol{\sigma}_\pi I_{INC} = N \tag{8}$$

Equation (8) can also be written in the form of the functional:

$$f_A \equiv f_A(\alpha, N) = N_{II} + \alpha N_\perp - N = 0 \tag{9}$$

Thus, to within small terms determined by the measurement statistics, the MPL calibration equations have the form:

$$f_J(\alpha, \varphi_{INC}, \delta_{INC}, \varphi_{SCA}, \delta_{SCA}, \xi) = 0 \tag{10}$$

$$f_A(\alpha, N) = 0 \tag{11}$$

## 3. Solution of the Calibration Equations

Consider the method of solving the MPL calibration equations. Note that the calibration parameters, $\Delta\varphi_{INC/SCA}$, define the initial angular positions of the wave plates. We consider that the wave plate rotation angles $\phi_{INC/SCA}$ in a series of states $\{\,\varphi_{INC},\,\varphi_{SCA}\,\}$ are accurate, where $\varphi_{INC/SCA} \equiv (\Delta\varphi + \phi)_{INC/SCA}$. In other words, $\phi_{INC/SCA}$ are known quantities included in the calibration equation as constants. We accept, as the available information, the Poisson statistics of the recorded signals [19]. We also consider that the measurement errors are random and independent. As customary, we use signals instead of averages and variances in a series of measurements. Recall that there are only three methods of estimating the parameters: the method of moments, the least squares method (LSM), and the maximum likelihood method [20]. Note that the regressors of the functionals (10) and (11) comprise the measurements $N_{II}$ and $N_\perp$; therefore, the residuals in fitting the calibration parameters depend on the measurements. The functional $f_A$ is linear in the parameters $\alpha$ and $N$, whereas the functional $f_J$ is nonlinear in the angular parameters.

In this case, the use of the LSM [20] is the optimal (economic) strategy. In addition, we consider the two-stage solution method:

1. The linear parameters $\widehat{\alpha}$ and $\widehat{N}$ are estimated as roots of the functional $f_A$;
2. Substitution of $\widehat{\alpha}$ into $f_J \rightarrow \widehat{f}_J$ is carried out;
3. The functional $\widehat{f}_J$ nonlinear in the angular parameters is solved with the Gauss–Newton method.

### 3.1. Stage 1 of Solving the Problem

The linear functional $f_A(\alpha, N) = 0$ is the classical model of pairwise regression with $m$ measurements:

$$y_i = \beta_1 x_i + \beta_2 + \varepsilon_i (i = 1, 2, \ldots, m) \tag{12}$$

in which $y_i$ is the model $y_i = -N_{II}$, and the regressors can be represented in the form of the row vector $\mathbf{A}_i = (x_i, -1) = (N_\perp, -1)$. The exogeneity of the regressors is violated since $\mathrm{cov}(y_i, \varepsilon_i) \neq 0$ and $\mathrm{cov}(x_i, \varepsilon_i) \neq 0$. Then, the LMS estimates are unbiased, but non-effective. The standard error estimates are biased and inconsistent. The problem is solved by using the generalized least squares (GLS) [21]. The system of Equations (12) in the matrix form has the form:

$$\mathbf{A}\boldsymbol{\beta} = \mathbf{Y} \tag{13}$$

where $\mathbf{A}$ is the $m \times 2$ matrix composed of the corresponding row vectors $\mathbf{A}_i$, $\boldsymbol{\beta} = (\beta_1, \beta_2)^T = (\alpha, N)^T$ is the column vector of the model coefficients, and $\mathbf{Y} = (y_1, y_2, \ldots, y_m)^T$ is the column vector.

The GLS model coefficients are estimated based on the Moore–Penrose pseudoinverse $\left(\mathbf{A}^T \mathbf{A}\right)^{-1} \mathbf{A}^T$ [22,23] as:

$$\widehat{\boldsymbol{\beta}} = \left(\mathbf{A}^T \widehat{\mathbf{D}}^{-1} \mathbf{A}\right)^{-1} \mathbf{A}^T \widehat{\mathbf{D}}^{-1} \mathbf{Y} \tag{14}$$

and the errors in the parameters $\widehat{D}\left[\widehat{N}\right]$ and $\widehat{D}[\widehat{\alpha}]$ are estimated as the corresponding diagonal matrix elements:

$$\left(\mathbf{A}^T \widehat{\mathbf{D}}^{-1} \mathbf{A}\right)^{-1} \tag{15}$$

The $m \times m$ error covariance matrix estimate $\widehat{\mathbf{D}} \equiv \widehat{\mathbf{D}}\left[\widehat{f}_A\right]$ in Equations (14) and (15) is defined as the diagonal matrix with elements:

$$\widehat{D}_{ii}\left[\widehat{f}_A\right] \cong N_{II} + \widehat{\alpha}^2 N_\perp \tag{16}$$

The estimates $\widehat{\boldsymbol{\beta}}$ are found using an iterative procedure because it is necessary to set the initial value $\alpha_0$ of $\widehat{\mathbf{D}}$. For the convenience of representation, the serial numbers of the iterations are omitted here. The subsequent estimates $\widehat{\boldsymbol{\beta}}$ are calculated from the previous estimates by solving Equations (14) and (15). The well-known convergence criterion is calculated from the residuals of the functional $\widehat{f}_A$. As a consequence of the Gauss–Markov theorem [24,25], the GLS vector estimate $\widehat{\boldsymbol{\beta}}$ is unbiased and effective. Since the matrix $\widehat{\mathbf{D}}$ is determined based on the available information, the GLS is often called the feasible GLS.

### 3.2. Stage 2 of Solving the Problem

In the next step (item 2) after the substitution, the functional $\widehat{f}_J \equiv \widehat{f}_J(\varphi_{INC}, \delta_{INC}, \varphi_{SCA}, \delta_{SCA}, \xi)$ takes the form:

$$\widehat{f}_J = f_0 - \widehat{c} = 0 \tag{17}$$

where $\widehat{c} = (N_{II} - \widehat{\alpha} N_\perp)/(N_{II} + \widehat{\alpha} N_\perp)$.

In the Gauss–Newton method, the nonlinear form $\widehat{f}_J$ in the first approximation of the Taylor series expansion is reduced to the linear form in the increments $\Delta$ of the calibration parameters:

$$\frac{\partial f_0}{\partial \varphi_{INC}}\Delta\varphi_{INC} + \frac{\partial f_0}{\partial \delta_{INC}}\Delta\delta_{INC} + \ldots + \frac{\partial f_0}{\partial \xi}\Delta\xi + \widehat{f}_J = 0 \tag{18}$$

where the partial derivatives of $f_0$, taking into account Equation (7), are determined by the subsequent substitution as the partial derivative components of the instrumental vectors given by Equations (A1)–(A10).

Let us consider Equation (18) as a model of multiple regression with $m$ measurements:

$$y_i = \Delta_1 x_i^{(1)} + \Delta_2 x_i^{(2)} + \ldots + \Delta_5 x_i^{(5)} + \varepsilon_i \, (i = 1, 2, \ldots, m) \tag{19}$$

where $y_i = -\widehat{f}_J$ is the model, the vector of the model coefficients $\mathbf{\Delta} = (\Delta_1, \Delta_2, \ldots, \Delta_5)^T$ is composed of the increments $\mathbf{\Delta} = (\Delta\varphi_{INC}, \Delta\delta_{INC}, \ldots, \Delta\xi)^T$, and the regressors in the form of the row vectors $\mathbf{J}_i = \left(x_i^{(1)}, x_i^{(2)}, \ldots, x_i^{(5)}\right)$ are composed of the corresponding partial derivatives $\partial f_0/\partial\varphi_{INC}$, $\partial f_0/\partial\delta_{INC}$, $\ldots$, $\partial f_0/\partial\xi$ given by Equations (A6)–(A10). In the matrix form, Equation (19) takes the following form:

$$\mathbf{J}\mathbf{\Delta} = \mathbf{Y} \tag{20}$$

where $\mathbf{J}$ is the $m \times 5$ matrix of the Jacobi derivatives composed of the row vectors $\mathbf{J}_i$ and $\mathbf{Y} = (y_1, y_2, \ldots, y_m)^T$. Here, Model (19) is endogenous with errors $\varepsilon_i$ that depend on measurements $\{N_{II}, N_\perp\}$. As in the preceding case, a solution can be found using the feasible GLS. The initial approximation vector is $\widehat{\mathbf{\beta}}^0 = \left(\varphi_{INC}^0, \delta_{INC}^0, \ldots, \xi^0\right)^T$. In the Gauss–Newton iterative method, the subsequent parameter estimates are related to the previous estimates through the increments:

$$\widehat{\mathbf{\beta}}^{j+1} = \widehat{\mathbf{\beta}}^j + \widehat{\mathbf{\Delta}}^j \, (j = 0, 1, 2, \ldots) \tag{21}$$

where $j$ is the serial iteration number. The GLS estimate of the coefficient increment vector $\widehat{\mathbf{\Delta}}^j$ has the form:

$$\widehat{\mathbf{\Delta}}^j = \left(\mathbf{J}^T \widehat{\mathbf{D}}^{-1} \mathbf{J}\right)^{-1} \mathbf{J}^T \widehat{\mathbf{D}}^{-1} \mathbf{Y} \tag{22}$$

where $\widehat{\mathbf{D}} \equiv \widehat{\mathbf{D}}\left[\widehat{f}_J\right]$ is the $m \times m$ covariance matrix of error estimates defined by the first terms of the Taylor series expansion:

$$\widehat{D}_{ii}\left[\widehat{f}_J\right] \cong \frac{N_{II} + \widehat{\alpha}^2 N_\perp + N_\perp^2 \widehat{D}[\widehat{\alpha}]}{\left(N_{II} + \widehat{\alpha}N_\perp\right)^2}\left(1 + \widehat{c}^2\right) \tag{23}$$

The errors of the parameters $\widehat{\mathbf{D}}\left[\widehat{\Delta}\right]$ are the corresponding diagonal elements of the matrix:

$$\left(\mathbf{J}^T \widehat{\mathbf{D}}^{-1} \mathbf{J}\right)^{-1} \tag{24}$$

The well-known convergence criterion is also based of the residuals of the functional $\widehat{f}_J$. The GLS estimate of the coefficient vector $\widehat{\mathbf{\beta}}$ is unbiased and effective.

Note the key point in compiling the error matrices in Formulas (16) and (23) of the available data:

- In stage 1 of the problem solution, the diagonal matrix $\widehat{\mathbf{D}}\left[\widehat{f}_A\right]$ is composed of the error estimates of measurements $N_{II}$ and $N_\perp$ as the initial data. As indicated above, taking

into account the properties of the Poisson statistics, they are the signals themselves $N_{II}$ and $N_{\perp}$;

- In the next stage, the diagonal matrix $\widehat{\mathbf{D}}\left[\widehat{f_J}\right]$ is composed of the error estimates for $N_{II}$, $N_{\perp}$, and $\widehat{\alpha}$ as the initial data; they are the signals $N_{II}$ and $N_{\perp}$ and $\widehat{D}[\widehat{\alpha}]$.

As shown in [20,26], if we denote the accuracy of estimates by $\widehat{\mathbf{D}}_{m=n}$ given that the number of unknowns coincides with the number of equations $m = n$, then the accuracy of the parameter estimates for $m > n$ is increased as $\widehat{\mathbf{D}}_{m \geq n} \cong \widehat{\mathbf{D}}_{m=n}/(m - n + 1)$.

## 4. Verification and Validation of the Calibration Method

The calibration method was verified for the MPL numerical model in which a series of measurements were performed with subsequent rotations of the transmitter and receiver $\lambda/4$ plates $R_{INC}$ and $R_{SCA}$, respectively. The method was validated on the results of MPL sensing of the Asian dust in Daejeon, Republic of Korea [11].

The convergence and stability of the method are critical to the set of angular positions of the transmission axes of the $\lambda/4$ plates { $\varphi_{INC}$, $\varphi_{SCA}$} in the measurement series below referred to as the 'set of positions' or simply the 'set'. To solve this problem, the search of the optimal variants of the set was performed. For this purpose, we first formed a set {$\varphi_{INC}$, $\varphi_{SCA}$}. Then for this set, we modeled $N$ instrumental vectors for the MPL receiver and transmitter as a set of states { $I_{INC}$, $I_{SCA}$}. Then, the corresponding series of signals were calculated from Formulas (3) and (4). Then the ranks, determinants, and condition numbers of the matrices $\mathbf{A}^T\mathbf{A}$ and $\mathbf{J}^T\mathbf{J}$ were calculated and compared. As the efficiency criterion, we used the condition of good conditionality of the systems of Equations (14) and (22). For simplicity, the sets were composed of placements with repetitions $A_n^m = n^m$ in two variants $A_3^2$ and $A_4^2$.

As a result, two optimal sets { $\varphi_{INC}$, $\varphi_{SCA}$} were chosen: the fast set with $A_3^2 = 9$ and the slow set with $A_4^2 = 16$. The fast set was { $\varphi_{INC/SCA}$ } = { $0, \pi3/8, \pi6/8$}, where the angles are in radians; the instrumental vector set { $I_{INC}$, $I_{SCA}$} for it was:

$$I_{INC/SCA} = \begin{pmatrix} * & * & * \\ 1 & 0.5 & 0 \\ 0 & -0.5 & 0 \\ 0 & 0.707 & -1 \end{pmatrix} \tag{25}$$

Here, the instrumental vectors are column vectors, and the asterisks are equal to 1 for $I_{INC}$ and {$-c$} for $I_{SCA}$.

The slow set was { $\varphi_{INC/SCA}$ } = { $0, \pi2/8, \pi5/8, \pi7/8$} with a periodicity of $\pi$ radians and the corresponding instrumental vector set { $I_{INC}$, $I_{SCA}$}:

$$I_{INC/SCA} = \begin{pmatrix} * & * & * & * \\ 1 & 0 & 0.5 & 0.5 \\ 0 & 0 & 0.5 & -0.5 \\ 0 & 1 & -0.707 & -0.707 \end{pmatrix} \tag{26}$$

### 4.1. Verification

Figures 2–4 show the results of the application of the calibration method to the MPL models with fast and slow sets { $\varphi_{INC}$, $\varphi_{SCA}$}. For convenience, we call these results the fast and slow estimates, respectively. Figure 2 shows the histograms of the fast and slow estimates of the parameters $\Delta\widehat{\delta}_{INC}$ (a) and $\Delta\widehat{\delta}_{SCA}$ (b) of the transmitter and receiver $\lambda/4$ plates, respectively. Here, $5 \times 10^4$:F is the fast estimate and $5 \times 10^4$:S is the slow estimate for <N> = $5 \times 10^4$; $5 \times 10^3$:F is the fast estimate and $5 \times 10^3$:S is the slow estimate for <N> = $5 \times 10^3$; 500:F is the fast estimate and 500:S is the slow estimate for <N> = 500 for both (a) and (b). The sample comprised 10,000 estimates of the MPL model

parameters. Hereinafter, the histograms are shown as linear envelopes of the cells for the best data visualization.

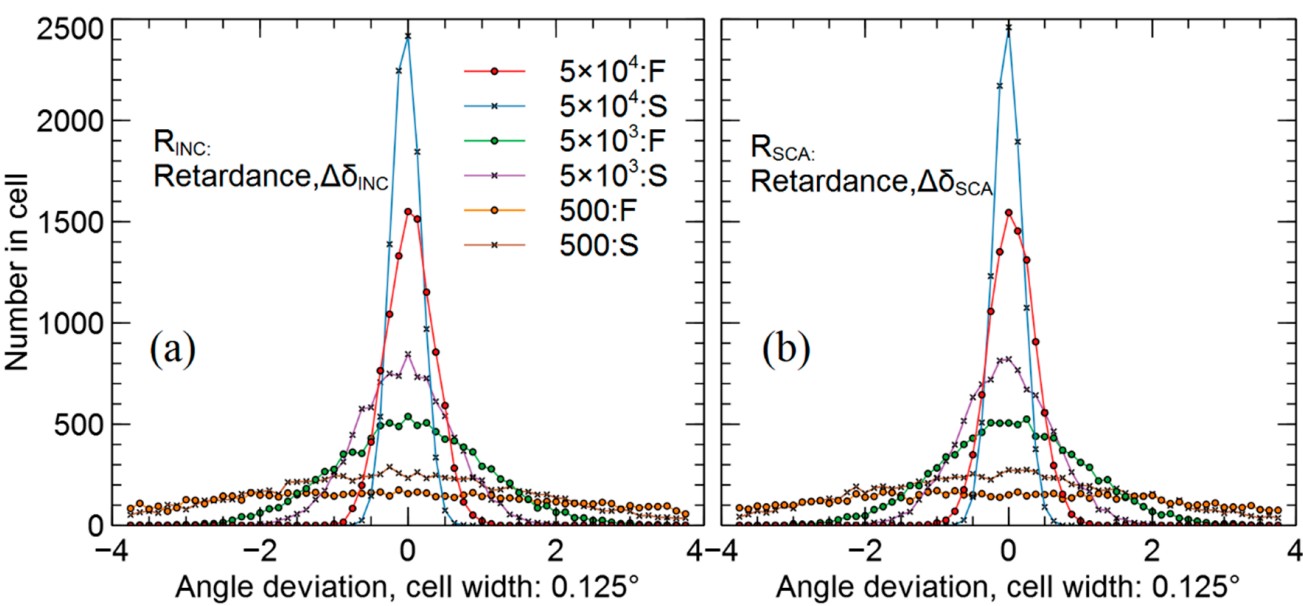

**Figure 2.** Histograms of the fast and slow estimates of the parameters $\Delta\widehat{\delta}_{INC}$ (**a**) and $\Delta\widehat{\delta}_{SCA}$ (**b**) of the $\lambda/4$ plates $R_{INC}$ and $R_{SCA}$. The designations for the fast and slow estimates are in the text.

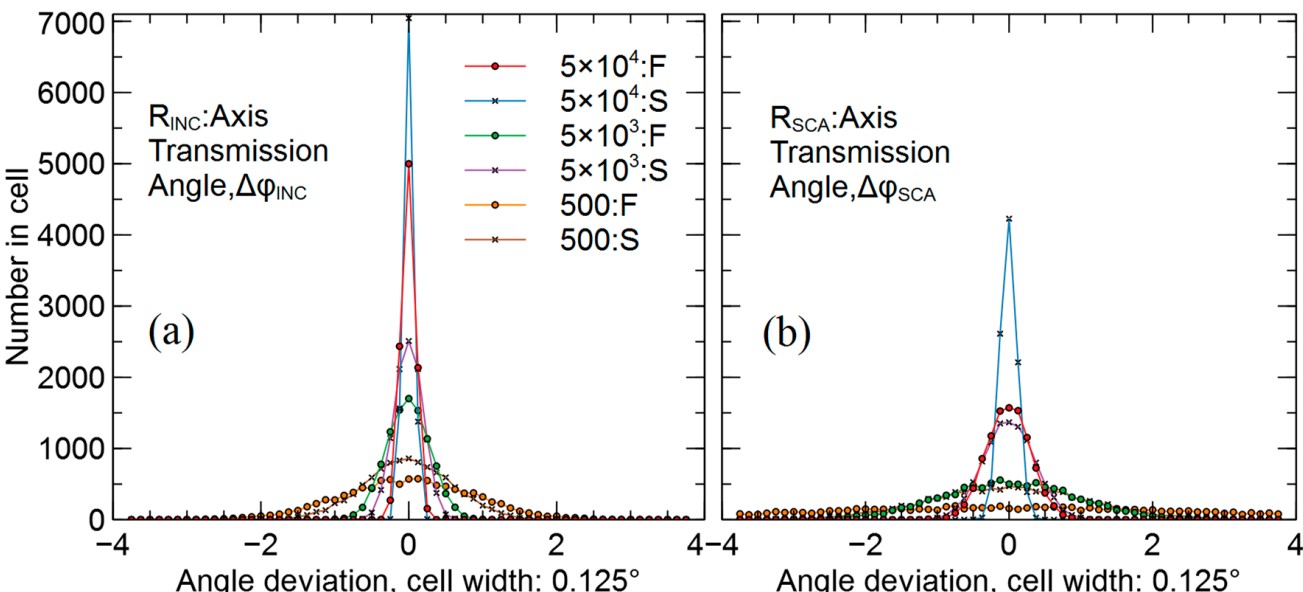

**Figure 3.** Histograms of the fast and slow estimates of the parameters $\Delta\widehat{\varphi}_{INC}$ (**a**) and $\Delta\widehat{\varphi}_{SCA}$ (**b**) of the $\lambda/4$ plates $R_{INC}$ and $R_{SCA}$. The designations for the fast and slow estimates are the same as in Figure 2.

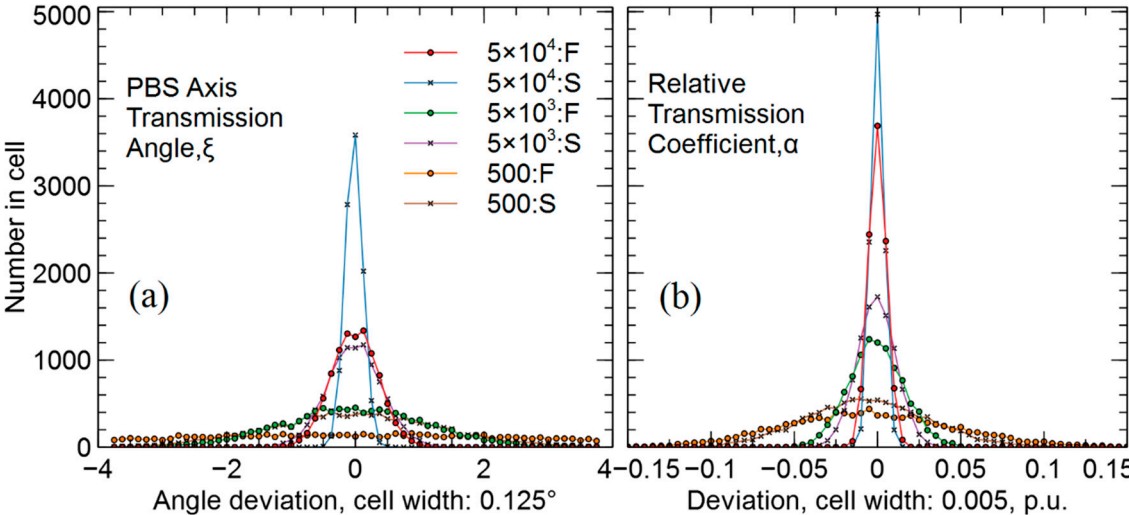

**Figure 4.** Histograms of the fast and slow estimates of the parameters $\widehat{\zeta}$ (**a**) and $\widehat{\alpha}$ (**b**). The designations of fast and slow estimates are the same as in Figure 2.

The following procedure of the verification of the calibration method was chosen. For convenience, it was assumed that the optical elements had no systematic angular position errors and $\gamma = 1$. For the fast and slow sets $\{\varphi_{INC}, \varphi_{SCA}\}$, a series of values $N$ was composed of the known average values $\langle N \rangle$: 100, 500, $10^3$, $5 \times 10^3$, $10^4$, $5 \times 10^4$. Then, the series of average signal values $\{\langle N_{II} \rangle, \langle N_\perp \rangle\}$ were calculated from Equations (3) and (4). Furthermore, the series of random echo-signals $\{N_{II}, N_\perp\}$ with the Poisson distribution were generated for the average signals. After that, the inverse problem of estimating the calibration parameters was solved. The initial approximation vector was set in the form $\widehat{\beta}^0 = (5°, 5°, 5°, 5°, 5°)$.

The estimation accuracy and the convergence of the method were estimated from statistical data. For this purpose, each procedure of generating a series $\{N_{II}, N_\perp\}$ and solving the inverse problem was repeated 10,000 times. Here, the calibration parameters are the initial angular positions of the $\lambda/4$ plates $\Delta\varphi_{INC}$ and $\Delta\varphi_{SCA}$, deviations from retardance of the $\lambda/4$ plates $\Delta\delta_{INC}$ and $\Delta\delta_{SCA}$, and the angle deviation $\zeta$ of the PBS.

Figure 3 shows the histograms of the angle deviations of the transmission axes estimates $\Delta\widehat{\varphi}_{INC}$ (a) and $\Delta\widehat{\varphi}_{SCA}$ (b) for the transmitter and receiver $\lambda/4$ plates, respectively.

Figure 4a shows the histograms of the angle deviations of the PBS transmission axis $\widehat{\zeta}$, and Figure 4b shows the histograms of the deviations of the relative transmission coefficient $\widehat{\alpha}$.

Table 1 presents the estimates of the average deviations of the calibration parameters for the examined sample of 10,000 results (marked in $\langle \rangle$ parentheses). The data are presented in columns of Table 1 depending on the signal-to-noise ratios (SNR).

**Table 1.** Estimations of the average deviations of the calibration parameters depending on the SNR.

| SNR | 223.6 | 100 | 70.7 | 31.6 | 22.4 | 10 |
|---|---|---|---|---|---|---|
| $\langle \widehat{\alpha} \rangle$ | $-0.00003 \pm 0.005$ $-0.00005 \pm 0.004$ | $-0.0003 \pm 0.01$ $-0.0003 \pm 0.008$ | $-0.0006 \pm 0.02$ $-0.0006 \pm 0.01$ | $-0.003 \pm 0.04$ $-0.003 \pm 0.03$ | $-0.005 \pm 0.05$ $-0.006 \pm 0.04$ | $-0.03 \pm 0.1$ $-0.03 \pm 0.08$ |
| $\langle \Delta\widehat{\delta}_{INC} \rangle$ | $0.03 \pm 0.3°$ $-0.03 \pm 0.2°$ | $0.05 \pm 0.7°$ $-0.04 \pm 0.5°$ | $0.04 \pm 1.0°$ $-0.05 \pm 0.6°$ | $-0.03 \pm 2.2°$ $-0.1 \pm 1.4°$ | $-0.2 \pm 3.1°$ $-0.2 \pm 1.9°$ | $-1.2 \pm 6.6°$ $-1.1 \pm 4.3°$ |
| $\langle \Delta\widehat{\varphi}_{INC} \rangle$ | $-0.007 \pm 0.09°$ $-0.003 \pm 0.06°$ | $-0.01 \pm 0.2°$ $-0.004 \pm 0.1°$ | $0.01 \pm 0.3°$ $-0.04 \pm 0.2°$ | $-0.01 \pm 0.6°$ $-0.003 \pm 0.4°$ | $-0.02 \pm 0.9°$ $-0.01 \pm 0.6°$ | $-0.02 \pm 1.9°$ $-0.04 \pm 1.3°$ |
| $\langle \Delta\widehat{\delta}_{SCA} \rangle$ | $0.05 \pm 0.3°$ $-0.01 \pm 0.2°$ | $0.06 \pm 0.7°$ $-0.03 \pm 0.4°$ | $0.06 \pm 1.0°$ $-0.03 \pm 0.6°$ | $-0.04 \pm 2.2°$ $-0.1 \pm 1.4°$ | $-0.1 \pm 3.1°$ $-0.2 \pm 2.0°$ | $-1.0 \pm 6.6°$ $-1.0 \pm 4.4°$ |

**Table 1.** *Cont.*

| SNR | 223.6 | 100 | 70.7 | 31.6 | 22.4 | 10 |
|---|---|---|---|---|---|---|
| $\langle \Delta\widehat{\varphi}_{SCA} \rangle$ | $-0.01 \pm 0.3°$ $-0.008 \pm 0.1°$ | $-0.02 \pm 0.6°$ $-0.007 \pm 0.3°$ | $-0.01 \pm 1.0°$ $-0.002 \pm 0.3°$ | $0.03 \pm 2.1°$ $-0.02 \pm 0.8°$ | $0.1 \pm 2.9°$ $-0.02 \pm 1.1°$ | $0.5 \pm 5.7°$ $0.04 \pm 2.5°$ |
| $\langle \widehat{\xi} \rangle$ | $-0.01 \pm 0.4°$ $-0.02 \pm 0.1°$ | $-0.01 \pm 0.8°$ $-0.02 \pm 0.3°$ | $-0.009 \pm 1.1°$ $-0.01 \pm 0.4°$ | $0.04 \pm 2.5°$ $-0.03 \pm 1.0°$ | $0.13 \pm 3.5°$ $-0.02 \pm 1.3°$ | $0.6 \pm 6.7°$ $0.07 \pm 2.9°$ |

The standard deviation (STD) of the signal served as the noise estimate. Here, $SNR = \langle N \rangle / \sqrt{\langle N \rangle}$ for the Poisson signal statistics. The upper figures in the table cells show the fast set average parameter estimates, and the lower figures show the slow set estimates.

The results demonstrate fast convergence to the functional minimum, on average, in two iterations and a good stability of the method for weak signals.

*4.2. Validation*

The proposed calibration method is scaled to the calibration parameters. This allowed us to validate the method using the data of polarization sensing in Daejeon, Republic of Korea, 36.34° N, 127.30° E on 1–2 June 2014 [11]. Figure 5 shows the vertical profiles of the scattering ratio estimates from the data of atmospheric sensing with the MPL. The scattering ratio profiles were calculated using the technique described in [8].

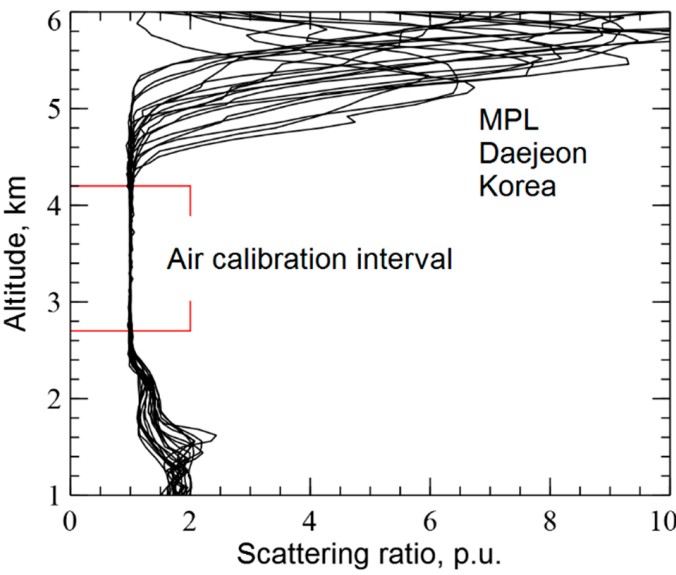

**Figure 5.** Vertical profiles of the scattering ratio. The boundaries of the MPL calibration range are indicated in red. MPL, Daejeon, Republic of Korea.

The matrix polarization lidar operated with a successive change of the receiver and transmitter $\lambda/2$ and $\lambda/4$ plates and without wave plates. These states are described as placement with repetitions $A_3^2 = 9$ [11]. There are six calibration parameters: the relative transmission coefficient $\alpha$; angular positions $\Delta\varphi_{INC}$ of the transmission axes of the $\lambda/2$ plate and $\Delta\psi_{INC}$ of the $\lambda/4$ plate of the receiver; $\Delta\varphi_{SCA}$ of the $\lambda/2$ plate and $\Delta\psi_{SCA}$ of the $\lambda/4$ plate of the transmitter; the angle $\xi$ of the PBS transmission axis. The vertical resolution of the measurements was 60 m. The altitude range from 2700 m to 4200 m with a prevalence of molecular scattering is clearly seen in Figure 5. The MPL was calibrated in this altitude range. Figure 6 shows the histograms of the signal estimates $\widehat{N}$. The characteristic shape of the histogram in Figure 6 is due to lidar echo signal variations as was shown by Equations (3) and (4).

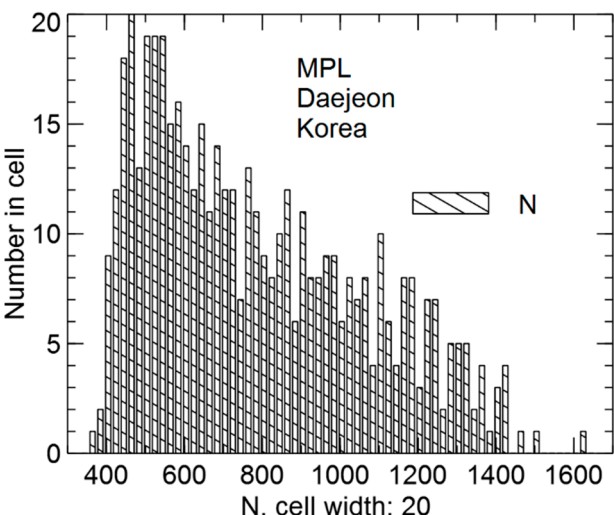

**Figure 6.** Histogram of the signal estimates $\widehat{N}$. MPL, Daejeon, Republic of Korea.

Here, the average signal estimate $\left\langle \widehat{N} \right\rangle$ = 784.26 with SNR = 28. The parameter estimates were obtained by the calibration method from the experimental data of 494 series of polarization measurements. Figure 7a shows the histograms of the angle deviation estimates $\Delta\widehat{\varphi}_{INC}$ and $\Delta\widehat{\psi}_{INC}$ of the transmission axes of the transmitter wave plates, the $\Delta\widehat{\varphi}_{SCA}$ and $\Delta\widehat{\psi}_{SCA}$ of the receiver wave plates, respectively, and of the PBS transmission axis $\widetilde{\xi}$. Figure 7b shows the histogram of the relative transmission coefficient $\widehat{\alpha}$.

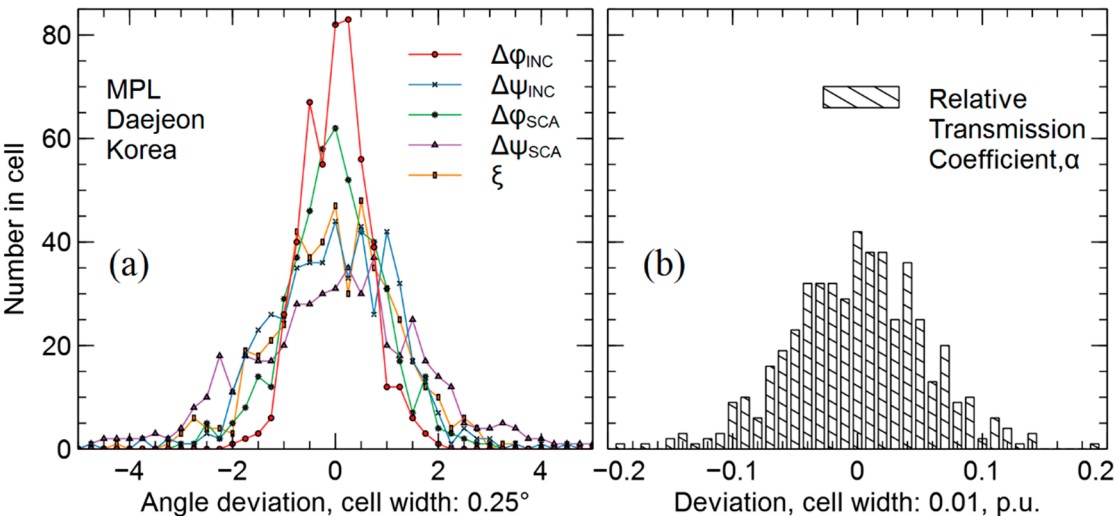

**Figure 7.** Histograms of the MPL parameter estimates, Daejeon, Republic of Korea. (**a**) Angle deviation estimates of the transmission axes of the transmitter ($\Delta\widehat{\varphi}_{INC}$ and $\Delta\widehat{\psi}_{INC}$) and receiver wave plates ($\Delta\widehat{\varphi}_{SCA}$ and $\Delta\widehat{\psi}_{SCA}$) and of the PBS $\widehat{\xi}$. (**b**) The deviation estimate of the relative transmission coefficient $\widehat{\alpha}$.

The histograms are shown relative to the MPL average calibration parameter estimates (marked in $\langle\ \rangle$ parentheses) given in Table 2.

**Table 2.** Average calibration parameter estimates.

| $\left\langle \widehat{\alpha} \right\rangle$ | $\left\langle \Delta\widehat{\varphi}_{INC} \right\rangle$ | $\left\langle \Delta\widehat{\psi}_{INC} \right\rangle$ | $\left\langle \Delta\widehat{\varphi}_{SCA} \right\rangle$ | $\left\langle \Delta\widehat{\psi}_{SCA} \right\rangle$ | $\left\langle \widehat{\xi} \right\rangle$ |
|---|---|---|---|---|---|
| $1.111 \pm 0.054$ | $-4.12 \pm 0.62°$ | $-1.8 \pm 1.2°$ | $-4.39 \pm 0.93°$ | $3.1 \pm 1.7°$ | $-2.7° \pm 1.2°$ |

Figure 8 shows the histogram of the convergence rate of the calibration method from the data of the outdoor experiment as the number of calibration procedures versus the number of iterations in the procedure. The sample histogram comprised 494 calibration procedures.

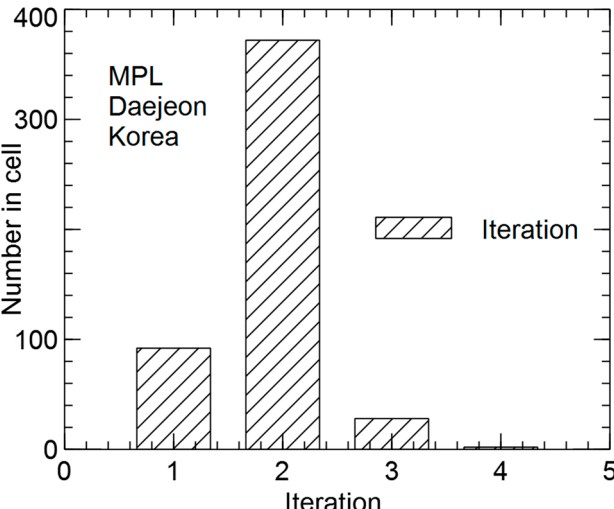

**Figure 8.** Histogram of the convergence rate of the MPL angular parameter calibration method, Daejeon, Republic of Korea.

On average, the convergence was reached in two iterations, which is characteristic for the least squares method. The results of the testing of the calibration method from the data of the outdoor experiment show a fast convergence and good stability of signals in the photoelectron counting mode.

## 5. Concluding Remarks

In this work, the principle of derivation and the method of solution of the equation of the MPL parameter calibration by air have been substantiated. To estimate the calibration parameters, the generalized least squares method was used. The minimum functional estimates obtained by this method are independent of the random variable distributions. The calibration parameter estimates are unbiased and effective.

The problem of the choice of the states of the wave plate set used for the calibration measurements was considered. In the mode of the MPL calibration by air, two sets of angular positions of the transmission axes of the lidar receiver and transmitter $\lambda/4$ plates in series of measurements were suggested.

The results of the method verification on the MPL model showed a fast convergence in 1–3 iterations and a stability of the parameter estimation method. The validation of the calibration method on the data of MPL sensing in Daejeon, Republic of Korea, confirmed the correctness of the chosen concept of method construction. This suggests the possibility of a practical application of the method of MPL calibration by air in polarization sensing.

It should be noted that in practice, the directions of the basis vectors $\mathbf{e}_z$ of optical MPL elements do not coincide with the z axis of the incident light beam. In addition, the MPL comprises other optical elements, for example, filters, lenses, etc., that introduce additional optical distortions. In the calibration by the proposed method, their contribution was taken into account in an implicit form; therefore, the MPL calibration parameter estimates should be considered effective. This is the advantage of the proposed calibration method compared to the correction technique.

Here, we have not considered the nature of the depolarization effect of the elements of the molecular scattering matrix; however, as shown in [27], the bandwidth of the input filters in the recording equipment should be taken into account. For spectral filter widths of less than 0.5 nm, filtering of the Raman scattering lines can lead to a change in the scattered radiation depolarization coefficient.

**Author Contributions:** Conceptualization, S.N.V., I.V.S. and D.-H.K.; methodology, S.N.V., I.V.S. and D.-H.K.; software, S.N.V., validation, S.N.V., I.V.S. and D.-H.K.; formal analysis, S.N.V., I.V.S. and D.-H.K.; investigation, S.N.V., I.V.S. and D.-H.K.; writing—original draft preparation, S.N.V.; writing—review and editing, S.N.V., I.V.S. and D.-H.K. All authors have read and agreed to the published version of the manuscript.

**Funding:** This work was supported by the IAO SB RAS State Assignment (No. 121031500341-3), by the Russian Science Foundation Grant No. 21-72-10089, and by the National Research Foundation of Korea (NRF) grant funded by the Korean government (MSIT) (No. 2020R1F1A1048293).

**Institutional Review Board Statement:** Not applicable.

**Informed Consent Statement:** Not applicable.

**Data Availability Statement:** Not applicable.

**Conflicts of Interest:** The authors declare no conflict of interest.

**Appendix A. Instrumental Vector**

Following [8,11], the MPL transmitter and receiver instrumental vectors $I_{INC}$ and $I_{SCA}$ can be represented in the form:

$$I_{INC} = (1, \, q_{INC}, \, u_{INC}, \, v_{INC})^T \tag{A1}$$

$$I_{SCA} = (-c, \, q_{SCA}, \, u_{SCA}, \, v_{SCA}) \tag{A2}$$

Here, the transmitter instrumental vector $I_{INC}$ is the Stokes vector parameter and $c$ is the polarization ratio as defined above. The components of the instrumental vectors (A1) and (A2) are defined as:

$$I_{INC} = (1, \, r_{22}, \, r_{32}, \, r_{42})^T \tag{A3}$$

$$I_{SCA} = (-c, \, Cr_{22} + Sr_{23}, \, Cr_{23} + Sr_{33}, \, Cr_{24} + Sr_{34}) \tag{A4}$$

Here, $S = \sin 2\xi$ and $C = \cos 2\xi$, the angle $\xi$ is the (least) angle of the PBS transmission axis, and $r_{ij}$ are the Mueller matrix elements of the wave plate. The Mueller matrix $\mathbf{M}(\varphi, \delta)$ of the transmitter and receiver wave plates has the following form [17]:

$$\begin{pmatrix} 1 & 0 & 0 & 0 \\ 0 & C^2 + S^2 \cos\delta & SC(1 - \cos\delta) & -S\sin\delta \\ 0 & SC(1 - \cos\delta) & S^2 + C^2 \cos\delta & C\sin\delta \\ 0 & S\sin\delta & -C\sin\delta & \cos\delta \end{pmatrix} \tag{A5}$$

where $S = \sin 2\varphi$ and $C = \cos 2\varphi$, $\varphi$ is the angle of the fast axis, and $\delta$ is the retardance.

For the partial derivatives $\partial I_{INC} / \partial \varphi_{INC}$ and $\partial I_{SCA} / \partial \varphi_{SCA}$, we have:

$$\frac{\partial I_{INC}}{\partial \varphi_{INC}} = (0, \, r_{22}, \, r_{32}, \, r_{42})^T \tag{A6}$$

$$\frac{\partial I_{SCA}}{\partial \varphi_{SCA}} = (0, \, Cr_{22} + Sr_{23}, \, Cr_{23} + Sr_{33}, \, Cr_{24} + Sr_{34}) \tag{A7}$$

The designations here coincide with those accepted above except that $r_{ij}$ here are the elements of the Mueller partial derivative matrix $\partial \mathbf{M}(\varphi, \delta) / \partial \varphi$ of the wave plate:

$$2 \times \begin{pmatrix} 0 & 0 & 0 & 0 \\ 0 & 2SC(\cos\delta - 1) & (C^2 - S^2)(1 - \cos\delta) & -C\sin\delta \\ 0 & (C^2 - S^2)(1 - \cos\delta) & 2SC(1 - \cos\delta) & -S\sin\delta \\ 0 & C\sin\delta & S\sin\delta & 0 \end{pmatrix} \tag{A8}$$

The formulas for the partial derivatives $\partial I_{INC}/\partial \delta_{INC}$ and $\partial I_{SCA}/\partial \delta_{SCA}$ coincide with those presented above except that $r_{ij}$ here are the elements of the Mueller partial derivative matrix $\partial \mathbf{M}(\varphi, \delta)/\partial \delta$ of the wave plate:

$$
\begin{pmatrix}
0 & 0 & 0 & 0 \\
0 & -S^2 \sin \delta & SC \sin \delta & -S \cos \delta \\
0 & SC \sin \delta & -C^2 \sin \delta & C \cos \delta \\
0 & S \sin \delta & -C \sin \delta & -\sin \delta
\end{pmatrix}
\tag{A9}
$$

For $\partial I_{SCA}/\partial \xi$, we obtain the expression:

$$
\frac{\partial I_{SCA}}{\partial \xi} = 2 \times (0, \; Cr_{23} - Sr_{22}, \; Cr_{33} - Sr_{23}, \; Cr_{34} - Sr_{24})
\tag{A10}
$$

where $r_{ij}$ are the elements of the Mueller wave plate matrix given by Equation (A5).

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
