# Peer review of "Calibration by Air in Polarization Sensing"

_atmosphere, doi:10.3390/atmos13081225_

Round 1
Reviewer 1 Report
General comments
The manuscript entitled with "Calibration by Air in Polarization Sensing", describes the theoretical framework for matrix polarization lidar (MPL) calibration. It presents a methodology for solving the calibration equations in a unbiased and effective way. Meanwhile, two calibration strategies (fast and slow sets) were proposed for MPL calibration and similar methodology was successfully applied in the calibration of a MPL in Daejeon, Republic of Korea.
The manuscript was well written. The methodology presented, can be very useful for characterizing systematic effects of MPL and conventional polarization lidars, which has been widely used in atmospheric remote sensing. I suggest acceptance after the authors address some minor issues.
1. A table summarizing all the important symbols, which were used in the analysis of lidar calibration, should be provided for better readability.
2. In terms of equation 2 for characterizing molecular phase function at backscatter angle, it's valid under the circumstance that the receiving module of the MPL can extract the Rayleigh scattering without any interference of rotation Raman spectrum. However, this can only be possible when very narrowband interference filter (< 0.5 nm) was used (Behrendt., et al, 2002). Even for modern optical techniques, it's still challenging or at least expensive to achieve this requirement. Therefore, I would like to see some discussions of the effects on the calibration results when interference filter with relatively large bandwidth (let's say FHWM ~1 nm) was used in the MPL.
3. page 3, line 100: here 'R_INC' and 'R_SCA' were used to denote the transmitter/receiver wave plate, but latter on, they were used to specify wave plate rotation angles (see page 4, line 150). Please correct it. And be consistent in the manuscript.
4. page 10, line 308: 'scattering ratio' should be explained here.
[1] Behrendt, A., and T. Nakamura (2002), Calculation of the calibration constant of polarization lidar and its dependency on atmospheric temperature, Opt Express, 10(16), 805-817.
Author Response
Response to the reviewer_1
Manuscript ID: atmosphere-1840597
Type of manuscript: Article
Title: Calibration by Air in Polarization Sensing
Authors: Sergei N. Volkov *, Ignatii V. Samokhvalov, Duk-Hyeon Kim
Dear reviewer_1,
Thank you for your interest in our publication. Thank you for the detailed analysis of our publication. Following your recommendations, we have made additions and corrections to the publications:
- a) In order to avoid interference in the designations of the angular positions of phase plates with the designations of phase plates, {R_INC, R_SCA} has been replaced everywhere in the text with {PHI_INC, PHI_SCA}. In addition, additional explanations to the accepted designations have been corrected and added;
- b) Following the logic of the accepted designations, explanations to the values in Tables 1 and 2 have been corrected and added;
- c) The corresponding symbols on the graphs have been corrected;
- d) Added a reference to the method used for calculating scattering profiles;
- e) In accordance with your recommendation, it is noted in section 5. Concluding Remarks that it is necessary to take into account the effect of the filter bandwidth on the values of the elements of the scattering matrix in the air;
- f) A corresponding reference has been added to the list of references.
With best Regards,
Sergei N. Volkov

Reviewer 2 Report
This paper describes a very interesting and robust procedure to provide critical calibration for polarized lidar systems able to extract microphysical properties of aerosol / smoke parameters. The paper is well organized and well written and is very detailed in it's analysis and I recommend acceptance of the paper.
One addition I would recommend to the authors to strengthen the interest of the paper to its readers is that it would be very interesting to see (or discuss the expected improvement) the effects of the calibrated system on retrieving the atmospheric profiles of specific atmospheric events in comparison to more traditional calibration correction techniques.
Author Response
Response to the reviewer_2
Manuscript ID: atmosphere-1840597
Type of manuscript: Article
Title: Calibration by Air in Polarization Sensing
Authors: Sergei N. Volkov *, Ignatii V. Samokhvalov, Duk-Hyeon Kim
Dear reviewer_2,
We thank you for a brief but exhaustive analysis of our publication. In accordance with your recommendations, we have carefully analyzed our publication and corrected the typos found. In addition, in order to present the material more clearly, we have refined the accepted designations for angular combinations and rotations of phase plates and calibration parameters.
With best Regards,
Sergei N. Volkov
